# Antioxidant Activity of Different Hop (*Humulus lupulus* L.) Genotypes

**DOI:** 10.3390/plants12193436

**Published:** 2023-09-29

**Authors:** Zala Kolenc, Tamara Hribernik, Tomaž Langerholc, Maša Pintarič, Maja Prevolnik Povše, Urban Bren

**Affiliations:** 1Laboratory of Physical Chemistry and Chemical Thermodynamics, Faculty of Chemistry and Chemical Engineering, University of Maribor, Smetanova ulica 17, SI-2000 Maribor, Slovenia; zala.kolenc@um.si; 2Department of Applied Natural Sciences, Faculty of Mathematics, Natural Sciences and Information Technologies, University of Primorska, Glagoljaška ulica 8, SI-6000 Koper, Slovenia; 3Department of Microbiology, Biochemistry, Molecular Biology and Biotechnology, Faculty of Agriculture and Life Sciences, University of Maribor, Pivola 10, SI-2311 Hoče, Slovenia; tomaz.langerholc@um.si (T.L.); masa.pintaric@um.si (M.P.); 4Department of Lifestock Breeding and Nutrition, Faculty of Agriculture and Life Sciences, University of Maribor, Pivola 10, SI-2311 Hoče, Slovenia; tamara.hribernik1996@gmail.com; 5Department of Animal Science, Faculty of Agriculture and Life Sciences, University of Maribor, Pivola 10, SI-2311 Hoče, Slovenia; maja.prevolnik@um.si; 6Institute of Environmental Protection and Sensors, Beloruska ulica 7, SI-2000 Maribor, Slovenia

**Keywords:** hop, antioxidant activity, FRAP, ORAC, intracellular antioxidative potential, genotypes

## Abstract

The antioxidant activity (AA) of hop extracts obtained from different hop genotypes (*n* = 14) was studied. For comparison, the purified β-acids-rich fraction and α-acids-with-β-acids-rich fraction were also used to test the antioxidative potential. The AA of purified hydroacetonic hop extracts was investigated using the Ferric Reducing Ability of Plasma (FRAP), Oxygen Radical Absorption Capacity (ORAC) and Intracellular Antioxidant (IA) methods. The FRAP values in different hop genotypes ranged between 63.5 and 101.6 μmol Trolox equivalent (TE)/g dry weight (DW), the ORAC values ranged between 1069 and 1910 μmol TE/g DW and IA potential values ranged between 52.7 and 118.0 mmol TE/g DW. Significant differences in AA between hop genotypes were observed with all three methods. AAs were determined using three different methods, which did not highly correlate with each other. We also did not find significant correlations between AA and different chemical components, which applies both to AA determined using individual methods as well as the total AA. Based on this fact, we assume that the synergistic or antagonistic effects between hop compounds have a more pronounced effect on AA than the presence and quantity of individual hop compounds.

## 1. Introduction

Antioxidants represent important health-promoting molecules, and their study is of great importance in food science, medicine and pharmacy [1]. Due to the ability of antioxidants to reduce oxidative stress in organisms, there are many studies supporting the fundamental role of antioxidants for human health and disease prevention/treatment [2,3,4]. The main mechanisms in assays measuring the antioxidant activity are based on hydrogen atom transfer (HAT) or single electron transfer (SET), reducing power, metal chelation, etc. [2,5]. Ferric reducing ability of plasma (FRAP) assay is based on the SET mechanism by the reduction of the complex of iron (III) to iron (II), which changes color when reduced [6,7]. Oxygen radical absorption capacity (ORAC) assay is based on the HAT mechanism by scavenging peroxyl radicals [6,8]. The methods used to determine the antioxidant activity (AA) are usually in vitro and chemical in their nature [9,10] although these methods do not realistically simulate the physiological environment (considering the entry of transmembrane molecules into cells, absorption with biomacromolecules and their interaction, bioavailability, toxicity and metabolism) [9]. For this purpose, the intracellular antioxidant (IA) potential has been developed [11,12,13], where the in vitro system in a cell may help to fully reveal the antioxidant capacity of plant extracts. Fluorescent dye 2’,7’-dichlorodihydrofluorescein diacetate (DCFH-DA) has been used to detect intracellular reactive oxygen species (ROS). DCFH-DA can freely pass through the cell membrane, but DCFH-DA itself does not exhibit fluorescence. Once DCFH-DA enters the cell, it is hydrolyzed by intracellular esterase to dichlorodihydrofluorescein (DCFH). Then, the oxidation of DCFH takes place by ROS. In this way, the fluorescent 2′,7′-dichlorofluorescein (DCF) is produced [12].

Numerous plant extracts have metabolites with more or less potent bioactive effects [14,15]. In this sense, there are a many scientific studies which explore the diverse biological effects of plants used in traditional medicine [16]. Due to the main application of hop in the beer industry, it was traditionally used for its anti-inflammatory, antimicrobial, diuretic, digestive, sedative, and progestogenic properties and also against insomnia [17]. Hop (*Humulus lupulus* L.) is recognized as a natural source of functional natural products with health effects [18,19]. In addition, hop natural products are becoming a potential source of bioactive compounds of great interest to the pharmaceutical, nutritional, veterinary and food industries [20]. Since hop forms a beer ingredient, the antioxidant properties of hop compounds are important in preventing the oxidation process during beer storage. Hop polyphenols are the most exposed compounds to oxidation and polymerization processes which result in beer turbidity [21].

The hop cones of a female plant are known for their use for beer flavoring and bittering purposes, since the 98% of the worlds hop production is applied in the beer industry [20]. Hop cones exert components such as resins, which are divided into hard resins (content in the range 3–5% of the total dry cone weight) and soft resins (content in the range of 10–25% of total dry cone weight) [22]. The soft resins are composed of two different bitter acid groups: α-acids (humulone, cohumulone, adhumulone, analogues) and β-acids (lupulone, colupulone, adlupulone, analogues) [18]. Depending on the hop variety, growing and climatic conditions, the content of α-acids and β-acids can vary greatly [18]. α-acids and β-acids are isomerized under high-temperature conditions to iso-α-acids and iso-β-acids. The iso-α-acids are considered the main contributors to beer bitterness, while the β-acids are held responsible for the characteristic beer taste [18,23]. The hop essential oils in the range of 0.5–3% are hop secondary metabolites placed in the lupulin. The hop polyphenols in the range of 4–14% of dry matter (DM) are further divided into flavanols, flavan-3-ols, phenolic carboxylic acids and other polyphenols [18,21]. Moreover, prenylflavonoids are a group of polyphenols, found only in the hop plant, with xanthohumol as their main representative [21] exhibiting numerous health effects [24,25]. 

To the best of our knowledge, there are no published studies on different worldwide hop genotype extracts (with known chemical composition of α- acids, β-acids and xanthohumol) and with related in vitro AA. Therefore, the aim of this study is to apply ORAC, FRAP and IA assays to determine the AA of different hop genotypes.

## 2. Results and Discussion

The importance of determining antioxidant properties has increased in recent decades due to the harmful role of free radicals present in foods [26]. In this work, the antioxidant potential was determined using three different methods: FRAP, ORAC and IA potential. The basic results of the determination of the AA are in Figure 1 and in Appendix A. Figure 1 depicts the comparison of AA between hop genotypes according to each method used in this study. The comparisons of hop varieties (and also β-acids-rich fraction (β-AF) as well as α-acids-and-β-acids-rich fraction (αβ-AF)) with respect to FRAP, ORAC and IA potential are displayed as box plots. 

The hop genotypes used in our study originated from different countries (Slovenia, Canada, Belgium, USA, Russia, Japan, England), but were all obtained from the hop gene bank maintained at the Slovenian Institute of Hop Research and Brewing in Žalec, Slovenia, as well as at the Hop Research Institute in Žatec, Czech Republic [27]. Based on the results presented in Figure 1, we can conclude that the use of hop extracts as a source of antioxidants is promising for the future due to their high AA. However, it is important to select the appropriate hop genotype for a specific purpose, as there are statistically significant differences between them in the case of all three AA determination methods (*p* < 0.05). The relationship between AA and the ageing stability of beer has already been observed [28,29]. To this end, and according to our results showing different AA with different hop genotypes, it is important to know which hop genotype is applied in the brewing process to achieve stability during beer aging with minimal use of food additives.

### 2.1. Antioxidant Activity Using FRAP

Among hop varieties, hydroacetonic extract (HAE) Savinjski golding had the highest FRAP value (101.6 μmol TE/g DW) and HAE Styrian Eureka exhibited the lowest FRAP value (63.5 μmol TE/g DW). Purified hop extract α-acids-and-β-acids-rich fraction (αβ-AF) had the overall highest FRAP value (121.3 μmol TE/g DW). High FRAP values were also found for HAE Caucasus S353 P15 (97.2 μmol TE/g DW), HAE Dekorativny (Russia) S248 (90.0 μmol TE/g DW), HAE Belgium S367 P157 (86.4 μmol TE/g DW), HAE Early promise (England) S68 (85.9 μmol TE/g DW), HAE Canada S353 P15 (82.8 μmol TE/g DW) and HAE Chocotsu No. 17 S168 Japan (82.0 μmol TE/g DW). In Figure 1, a significant difference (*p* < 0.05) was observed between different hop genotypes in FRAP values. The reducing activity of FRAP was also measured in the study by Abram et al. [30], where significant differences were found between the varieties Magnum and Aurora, when the reducing activity of FRAP (expressed as the mean slope linear regression at different concentrations) was determined in ethanolic hop cone and hop leaf extracts across three different years. In a recent study, the average FRAP value in extracts from ethanolic macerated dried hop pellets was determined as 4.12 mg ascorbic acid equivalent/g DW [26]. However, the highest FRAP value was determined in αβ-AF, where the highest α-acids content was determined (12.76%; *w*/*w* of cohumulone, 72.79%; *w*/*w* of n+adhumulone) [27].

### 2.2. Antioxidant Potential Using ORAC

In the data obtained from the ORAC assay, HAE Styrian Eureka (1910 μmol TE/g DW) had the highest ORAC value and HAE Aurora exhibited a similarly high ORAC value (1909.1 μmol TE/g DW). High ORAC values were also found for HAE Belgium S367 P157 (1735.8 μmol TE/g DW), HAE Nugget (USA) S222 (1723.8 μmol TE/g DW), HAE Styrian Fox (1722.3 μmol TE/g DW), HAE Savinjski golding (1695.6 μmol TE/g DW), HAE Styrian Wolf (1686.7 μmol TE/g DW), HAE Dekorativny (Russia) S248 (1686.9 μmol TE/g DW) and HAE Styrian Eagle (1653.7 μmol TE/g DW), while HAE Caucasus S353 P15 extracts exhibited the lowest ORAC value (1069 μmol TE/g DW). In Figure 1, a significant difference (*p* < 0.05) was observed between different hop genotypes in ORAC values. The lowest ORAC value was found for αβ-AF. To the best of our knowledge, there are no published studies using the ORAC method to determine the AA of hop extracts. The ORAC method has previously been used to determine the AA for individual hop components such as humulones, lupulones, isohumulones, reduced isohumulones, tetrahydro-isohumulones or xanthohumol [31]. Moreover, the ORAC method was also applied to determine AA in beer [6]. Other studies revealed significant differences between different hop genotypes in AA with 2,2-diphenyl-1-picrylhydrazyl (DPPH) assay [32] and DPPH, FRAP assay [30], as well as DPPH, 2,2′-azino-bis-(3-ethylbenzothiazoline-6-sulfonic acid) (ABTS) assay [23].

### 2.3. IA Potential Determination

#### 2.3.1. Cytotoxicity Assay

The data obtained from the cytotoxicity assay showed that dimethyl sulfoxide (DMSO) in different concentrations (1.0%, 0.5%, 0.25%, 0.125%) does not exhibit toxic effects on cell metabolic activity. In this way, the toxic effect of the solvent was excluded.

Different concentrations of hop HAE were also used in the cytotoxicity assay (125.00 µg/mL, 62.50 µg/mL, 31.25 µg/mL, 15.63 µg/mL, 7.81 µg/mL, 3.91 µg/mL, 1.95 µg/mL, 0.98 µg/mL). The obtained data showed that different genotypes of hop samples and a sample of β-AF do not exert a cytotoxic effect at concentrations ≤125.00 µg/mL. The sample of αβ-AF does not exhibit a cytotoxic effect at concentrations ≤250.00 µg/mL.

#### 2.3.2. IA Potential Assay

Our working concentration of different hop genotypes and of αβ-AF samples was 6.25 µg/mL and 3.1 µg/mL for the β-AF sample. As presented in Figure 1, the highest IA potential among hop varieties was determined for HAE Nugget (USA) S222 (118.0 mmol TE/g DW). High IA potentials were also measured in HAE Styrian Eagle (112.4 mmol TE/g DW) and HAE Dekorativny (Russia) S248 (106.2 mmol TE/g DW) varieties. The lowest IA potential was exhibited by HAE of Savinjski Golding (52.7 mmol TE/g DW) and HAE Caucasus S353 P15 (65.0 mmol TE/g DW). Among IA potential data, a significant difference (*p* < 0.05) was again observed between different hop genotypes (Figure 1). The greatest IA potential was obtained for β-AF (150.4 mmol TE/g DW). According to the scientific literature, overall, there is no study available on the comparison of IA potential among different hop genotypes. In a recent study [33], the IA potential was determined in a hop extract (variety not specified) using DCFH-DA. The effect of IA potential at a concentration of 32.00 µg/mL was compared to the potent antioxidant flavonoid luteolin. The IA potential of hop was measured in another study [34], where leaves of *Humulus japonicus* were applied for the preparation of hop extract and the extract showed a significant reduction in reactive oxygen species (ROS) compared to control cells without the extract.

### 2.4. Correlation between AA with Different AA Methods 

The Spearman rank correlation was subsequently calculated between all three methods for the antioxidative property determination of hop extracts (Table 1). The results showed a positive, but not statistically significant, correlation between FRAP and IA methods (0.359) as well as between ORAC and IA methods (0.227), while ORAC was significantly negatively correlated with FRAP (−0.511). The correlation between different methods for AA determination in our study was therefore low. This is consistent with another study on different genotypes of vegetables such as cabbage, carrot, cauliflower, onion, tomato and pepper [35]. On the contrary, there is a similar study on 18 different onion genotypes [36] with the comparison of FRAP and DPPH methods reporting a good correlation between them.

### 2.5. Correlation between AA and Chemical Composition of Different Hop Genotypes

The Spearman rank correlation was determined first between methods for the AA determination of hop extracts and their chemical composition (Table 2). The FRAP method results are negatively correlated with all determined hop chemical compounds and are highly positive correlated (0.656) with undetermined substances of the HAE, indicating that substances other than the identified ones are responsible for high FRAP values. The AA determined using the ORAC method was positively correlated with xanthohumol (0.434), cohumulone (0.124), n+adhumulone (0.102) and unknown substances (0.175), while on the other hand negatively correlated with colupulone (−0.166) and n+adlupulone (−0.184), which are representative of β-acids (−0.219). Finally, the IA potential method indicated a positive correlation with xahthohumol (0.044), colupulone (0.137) and n+adlupulone (0.159), which are representative of β-acids (0.109). On the other hand, a negative correlation was obtained between IA potential and cohumulone (−0.129) and n+adhumulone (−0.268) which are representative of α-acids (−0.271). It is important to note that the obtained correlations were statistically not significant, indicating that antioxidant activity cannot be attributed to specific substances, but rather to a more complex relationship between the components.

### 2.6. Association of AA with Chemical Composition

The chemical composition of hop extracts of different genotypes was presented in our previous study [27]. Hereafter, these results were used to compare the AA of these extracts with their chemical composition. To the best of our knowledge, there is no published study that determines the AA of hop extracts with respect to their chemical composition as is presented in the results obtained in this study (Table 3). 

The quantities of chemical compounds’ individual classes determined using the different AA methods are presented in Table 3. The hop samples were divided into three classes (low, medium and high AA), with Q1 and Q3 of individual AA methods as the limits among classes. In the determination of AA, the FRAP method contained the highest value of n+adhumulone (28.45%, *w*/*w*) in the low class. AA determined using the FRAP also revealed the highest xanthohumol content in the low class (1.87%, *w*/*w*), the highest cohumulone content (5.31%, *w*/*w*) in the high class, and the highest colupulone content (6.85%, *w*/*w*) and the highest n+adlupulone content (8.59%, *w*/*w*) in the medium class. Moreover, the lowest unknown content (48.1%, *w*/*w*) part of the hop extracts was observed in the low class, determined using FRAP.

AA determination using the ORAC revealed the highest xanthohumol content (1.45%, *w*/*w*) in the high class, which is very close to the medium class (1.44%, *w*/*w*), the highest cohumulone content (6.79%, *w*/*w*) in the medium class, the highest n+adhumulone content (22.59%, *w*/*w*), cohumulone (8.13%, *w*/*w*) and n+adlupulone (9.47%, *w*/*w*) contents in the low class. On the other hand, the lowest unknown part of the hop extracts was found in the low class (54.7%) and in the medium class (57.9%).

The determination of AA using the IA potential method disclosed the highest xanthohumol (1.52%, *w*/*w*) and cohumulone (6.72%, *w*/*w*) content in the medium class, the highest n+adlupulone content (26.04%, *w*/*w*) in the low class, and the highest colupulone content (7.23%, *w*/*w*) and n+adlupulone (8.35%, *w*/*w*) contents in the high class. On the other hand, the lowest unknown part of the hop extracts was observed in the low class (55.4%).

To sum up the results of AA class comparison with respect to the FRAP, ORAC and IA potential methods individually, the total AA class was also constructed based on all three AAs. It was observed that the individual identified hop compounds do not have a large effect on different AA values. Therefore, it could be concluded that the basic chemical composition of different hop genotype extracts does not have a great influence on the AA. From the fact that there is a poor correlation of AA also with the unknown part of the hop extracts (R = 0.141 for IA, R = 0.175 for ORAC and R = 0.656 for FRAP method), it cannot be concluded that the hop compounds not identified in this study are important for the AA of the hop extracts. It has been reported that the synergistic or antagonistic effects between different hop compounds may be responsible for the results obtained in this study. Namely, a recent review [17] was published on hop bioactive compounds and concluded that hop compounds act synergistically and combine their antioxidant effects. In a study in which AA was determined using ABTS and DPPH assays, although the chemical composition of ethanolic hop extracts was reported in the paper, there was no intention to correlate the chemical composition of hop extracts with the obtained AA in the different hop genotypes expressed [23]. Nevertheless, the hop extracts and individual hop compounds exert numerous biological effects due to their high AA [37]. By using the inhibition (EC_50_ was determined) of the hydroxyl radical method, the total AA (based on the protection of the β-carotene-linoleic acid model system) was determined for α-acids, β-acids, dihydro-iso-α-acids, tetrahydro-iso-α-acids, hexahydro-iso-α-acids, hexahydro-iso-β-acids and hop oil; as a result, the α-acids elicit a higher AA than β-acids [38]. The comparison of five different hop genotypes (Sa-1, Nugget, Chinook, Marco Polo and Tsingdao) was also performed in a previously mentioned study, in which the Marco Polo showed the highest antioxidant capacity and hydroxyl radical scavenging activity [38]. Moreover, it was reported that the hop genotype Cascade exerts diverse AA (determined by the DPPH and ABTS assays) according to the different growing area (Brasilia in comparison to USA), indicating the importance of environment for the obtained AA values [39].

Considering the standardization of the AA results and the formation of total AA classes based on normalization (Figure 2), it was found that the highest xanthohumol content was determined in the low AA class (1.69%, *w*/*w*). The content of hop α-acids is similar in the low and medium AA classes, namely, the content of cohumulone in the low AA class is 6.59%, *w*/*w* and in the medium AA class is 6.30%, *w*/*w*. Similarly, the content of n+adhumulone in the low AA class is 21.72%, *w*/*w* and in the medium AA class is 23.38%, *w*/*w*. On the other hand, the content of β-acids is higher in the low AA class than in the medium AA class. Thus, the colupulone content in the low AA class is 6.94%, *w*/*w*, while it is 4.29%, *w*/*w* in the medium AA class. Similarly, the n+adlupulone content in the low class is 9.40%, *w*/*w* and it is lower (3.96%, *w*/*w*) in the medium AA class. However, the content of β-acids was also significant in the high AA class. The unidentified hop extract content was the lowest in the low AA class (53.66%, *w*/*w*), and higher in the medium (61.09%, *w*/*w*) as well as in the high AA class (63.86%, *w*/*w*).

### 2.7. Principal Component Analysis (PCA) of Different Hop Genotype Samples according to AA

In this study, the PCA has also been performed to reduce the number of variables in a dataset while retaining the most important information. Principal components (PC) are derived to explain the variance in the data. Promax with Kaiser normalization was applied as the rotation method to extract two PCs. In Table 4, the PC 1 is highly loaded by the identified chemical compounds, i.e., xanthohumol (0.907), cohumulone (0.819), n+adhumulone (0.843), colupulone (0.993) and n+adlupulone (0.882). Not surprisingly, the unknown part of the hop extracts has a negative load in PC1 (−0.948), since a higher proportion of identified substances is naturally followed by a lower rate of unknown substances. The PC 2 was highly loaded by AA methods ORAC (0.814) and IA potential (0.768) while negatively by FRAP (−0.745). PC2 suggests that the ORAC and IA methods do correlate with each other, explaining variance in the system, while FRAP results have the opposite effect. 

## 3. Materials and Methods

### 3.1. Hop Extract Preparation and HPLC Determination of Chemical Composition

Fourteen different hop genotypes were included in the determination of their AA. The hop cones were obtained from the hop gene bank maintained at the Slovenian Institute of Hop Research and Brewing as well as at the Hop Research Institute, Czech Republic, during the 2019 harvest season. In addition, the β-AF and the combination of αβ-AF, obtained from Hopsteiner (Mainburg, Germany) and from Labor Veritas (Zürich, Switzerland), respectively, were included in AA determination. The hop extract preparation was performed by using acetone–water (9:1) as solvent by following the methodology of Kolenc et al. [27] and the HAEs of different hop genotypes were obtained. The samples applied in this study were exactly the same as in a previous antimicrobial study [27]. Moreover, the chemical composition data (performed at the Slovenian Institute of Hop Research and Brewing) used for the statistical analyses in this study (Figure 3) were also obtained in this previous study authored by Kolenc et al. [27].

### 3.2. Antioxidant Activity Using FRAP

#### 3.2.1. Chemicals

Analytical grade 2,3,5-triphenyl tetrazolium chloride (TPTZ), dimethyl sulfoxide (DMSO) and Trolox were purchased from Sigma Aldrich.

#### 3.2.2. FRAP Assay Method

The FRAP assay was performed using 300 mM acetate buffer (pH = 3.6). The FRAP working reagent consists of acetate buffer (300 mM, pH = 3.6), TPTZ (10 mM dissolved in 40 mM HCl) and 20 mM iron (III)-chloride solution (prepared in water) in the ratio 10/1/1 (*v*/*v*/*v*). The AA was determined by applying 96-well plates and a micro-plate reader (Tecan, Männedorf, Zürich, Switzerland). The stock solutions of samples (HAE, αβ-AF and β-AF)/Trolox were prepared in DMSO, whereas the working solutions were prepared in 300 mM acetate buffer (125 µg/mL was the final concentration in the plate for both). For the HAE, αβ-AF and β-AF, six two-fold dilutions were prepared for each sample and the concentration that fits the middle of the Trolox calibration curve was used as a result. The final well volume in the plate was 200 µL; for this purpose, 100 µL of the FRAP working reagent was added as well as 100 µL of the diluted (in acetate buffer) sample/Trolox (for calibration curve) or water (for blank). The 40 mM stock solution of Trolox (positive control) in DMSO was applied to prepare the calibration curve. The Trolox solution was then diluted to the desired concentration (0.5–62.5 µM) by using the acetate buffer (300 mM, pH = 3.6). The complete reaction mixture was stirred and incubated at room temperature for 10 min. The absorbance was measured at 593 nm by a micro-plate reader (Tecan, Männedorf, Zürich, Switzerland). The result was expressed as Trolox equivalents per gram of dry weight of the sample (µmol TE/g DW). Samples were measured in quadruplicates, and the mean and standard deviation were calculated. The data were analyzed by using the Microsoft Excel program. The FRAP assay was performed, with minor modifications, as per Wannenmacher et al. [6].

### 3.3. Antioxidant Activity Using ORAC

#### 3.3.1. Chemicals

Fluorescein (3′,6′-dihydroxyspiro[isobenzofuran-1(3H),9′-[9H]xanthen]-3-one), AAPH (2,2’-azobisisobutyramidinium chloride), DMSO and Trolox were purchased from Sigma Aldrich and were analytical grade.

#### 3.3.2. ORAC Assay Method

The ORAC assay was performed with minor modifications by [40]. The fluorescent probe in our experiment was the fluorescein. AAPH was used as a peroxy radical generator. The AA was determined using 96-well black plates and a micro-plate reader (Tecan, Männedorf, Zürich, Switzerland). To avoid the marginal and temperature effect, only the inner 60 wells were applied for experimental purposes. The outer wells were filled with 200 µL of distilled water (200 µL was again the final well volume). The 40 mM stock solution of Trolox (positive control) in DMSO was used to prepare the calibration curve. The Trolox solution was then diluted to the desired final concentration on plates (0.39–12.5 µM) using the phosphate buffer (75 mM, pH = 7.4). Fluorescein was prepared as a 30 mM stock solution in DMSO. The working fluorescein solution was diluted with phosphate buffer (75 mM, pH = 7.4) to the final concentration of 150 nM. The stock solutions of samples (HAE, αβ-AF and β-AF) were prepared in DMSO, whereas the working solutions were prepared in 75 mM phosphate buffer (1 µg/mL was the final concentration on a plate for all). For the HAE, αβ-AF and β-AF, six two-fold dilutions were prepared for each sample and the concentration that fits the middle of the Trolox calibration curve was used as a result. The final reaction volume was 200 µL in a 75 mM phosphate buffer (pH = 7.4). Sample/Trolox was placed on the plate in a volume of 25 µL and the fluorescein in a volume of 150 µL. The plate was then incubated for 10 min at 37 °C with gentle stirring. The AAPH was freshly prepared in phosphate buffer at 15 mM concentration and quickly added to the plate reaction mixtures in a volume of 25 µL. The microplate was immediately placed in the microplate reader and the fluorescence was recorded every cycle for 90 cycles with a 485 nm excitation filter and 528 nm emission filter. The final ORAC values were calculated according to the procedure described by Huang et al. [40]. To sum the calculation up, the net area under the curve (AUC) was calculated for both Trolox standards and samples using Equation (1):(1)$$AUC=0.5+\left(\frac{f_{2}}{f_{1}}\right)+\left(\frac{f_{3}}{f_{1}}\right)+\left(\frac{f_{4}}{f_{1}}\right)+\ldots +0.5\;(\frac{f_{90}}{f_{1}})$$
where the f_1_ represents the initial fluorescence read at the first cycle, the f_2_ represents the fluorescence, which was read at the second cycle, … and the f_90_ represents the last ninetieth-read fluorescence. For each sample/Trolox standard, the net AUC was calculated by subtracting the blank sample. The final ORAC values were calculated by using the regression equation between Trolox concentration and the net AUC of each Trolox standard. The result was expressed as Trolox equivalents per gram of dry weight of the sample (µmol TE/g DW). Samples were measured in triplicate, and the mean and the standard deviation were calculated. The data were analyzed using the Microsoft Excel program.

### 3.4. Determination of IA Potential

#### 3.4.1. Chemicals

Dulbecco’s Modified Eagle Medium (DMEM) and Fetal Bovine Serum (FBS) were obtained from Gibco (Paisley, UK). L-glutamine was purchased from Life Technologies. Penicillin, streptomycin, 2’,7’-dichlorodihydrofluorescein diacetate (DCFH-DA), tert-butyl hydroperoxide (t-BuOOH), 6-hydroxy-2,5,7,8-tetramethylchroman-2-carboxylic acid (Trolox), 3-(4,5-dimethylthiazol-2-yl)-2,5 diphenyl tetrazolium bromide (MTT) and DMSO were purchased from Sigma Aldrich.

#### 3.4.2. Cell Culture

Human non-carcinogenic macrophage cell line TLT was used for the cytotoxicity assay and for the determination of IA potential [41]. The cell passage number was kept below 30 in all experiments. Cells were cultivated in DMEM, which was supplemented with 5% FBS, 2 mM L-glutamine and with two antibiotics: 100 IU/mL penicillin and 0.1 mg/mL streptomycin. Cells were cultivated in T-culture flasks with a surface of 25 cm^2^ in an incubator with a humidified atmosphere, at a constant temperature of 37 °C and in 5% CO_2_. The medium was changed as needed and the cells were subcultured when they reached confluence. 

#### 3.4.3. Cytotoxicity Assay

Before the IA potential was determined, the cytotoxicity test was carried out. Using the cytotoxicity assay, the nontoxic concentration of hop extracts was determined. To assess cell viability, the MTT assay was used. The MTT assay was used, with slight changes, as per Mosmann [42]. Cells were seeded in a transparent 96-well microtiter plate at a density of 5 × 10^4^ cells per each well in a humidified atmosphere with 5% CO_2_ at 37 °C. After 24 h of incubation, cell media were removed, and different concentrations of HAE were added. Stock solutions of hop extracts (100 mg/mL) were prepared in DMSO and different concentrations of hop extracts were also prepared in DMEM. Additionally, different concentrations of DMSO were tested; to exclude the toxic effect of DMSO, this served as a negative control. High concentrations of hop extracts (up to 1 mg/mL) served as a positive control. The plate was incubated for 2 h. After the incubation, the stock solution of MTT was prepared in DMEM (5 mg/mL) and MTT was added into each well, with a final concentration of 0.5 mg/mL. The plate was incubated at a room temperature for 5 min with stirring and then put in the incubator for another two hours, so that the formazan crystals were formed. After the incubation, the medium with MTT was removed and the plate was dried. Afterwards, 100 µL of 0.04% HCl in isopropanol was added into each well. The plate was incubated for 15 min at a room temperature and then the absorbance was measured at 570 nm and 630 nm with a microplate reader (Tecan, Männedorf, Zürich, Switzerland). Data were analyzed using the Microsoft Excel program. The final absorbance was calculated from the difference between the absorbance measurements at 630 and 570 nm. Absorbance values of hop extracts and DMSO concentrations were expressed in percentages, considering values of absorbances in the control wells, which represented 100%.

#### 3.4.4. IA Potential

For the determination of the IA potential of the hop extract samples (HAE, αβ-AF and β-AF), DCFH-DA was used. The protocol of Girard-Lalancette et al. [11] was used, with slight changes. Cells were seeded in black 96-well microtiter plates in a humidified atmosphere with 5% CO_2_ at 37 °C at a density of 1.5 × 10^4^ cells per well, except in the edge wells of the plate. In this way, the edge marginal effect was excluded. After 24 h of incubation, the cell media were removed, and cells were washed with Hank’s Balanced Salt Solution (HBSS). Trolox was used as a standard and a positive control in different concentrations (800 µM, 700 µM, 600 µM, 500 µM, 400 µM, 300 µM) and investigated hop samples were added to the microtiter plate. The negative control used in our experiment was HBSS without added substances. Stock solutions of Trolox (40 mM) and hop extracts (100 mg/mL) were prepared in DMSO, whereas the working solutions were prepared in the HBSS. Each concentration of Trolox and each hop extract was tested in five replicates. The plate with Trolox and hop extracts was incubated for two hours in a humidified atmosphere with 5% CO_2_ at 37 °C. After incubation, the cells were washed with HBSS, and then incubated with 20 µM DCFH-DA for another 30 min. The stock solution of DCFH-DA (33 mM) was prepared in DMSO, whereas the final concentration of DCFH-DA (20 µM) was prepared in HBSS. Again, the cells were washed with HBSS and then treated with 100 µM tert-butyl hydroperoxide (t-BuOOH) in HBSS. After 2.5 h of incubation with 5% CO_2_ at 37 °C, the fluorescence was measured with a microplate reader (Tecan, Männedorf, Zürich, Switzerland). The excitation wavelength was set at 485 nm and the emission wavelength at 528 nm. The results were expressed as Trolox equivalent per gram of dry weight (mmol TE/g DW).

### 3.5. Statistical Analysis

All statistical analyses were performed using SPSS Statistic (IBM, version 28, Armonk, NY, USA).

Initially, the Shapiro–Wilk test was used to test the normal distribution of all three AA assays (FRAP, ORAC and IAP) in total and within each hop variety. In the next step, Levene’s test was applied to test the equality of variances among the studied hop varieties for all three AA assays. As both tests proved the normal distribution of variables, parametric statistics were applied for subsequent analyses. They included (i) the comparison of hop varieties regarding AA, (ii) an analysis of the relationship between different AA methods and (iii) an analysis of AAs relationship with studied chemical compounds. 

The effect of hop variety on AAs was calculated using ANOVA. In the case of an applied statistically significant effect (*p* < 0.05), a post hoc Duncan’s multiple range test was further used to test differences among hop varieties. The comparisons of hop varieties with respect to FRAP, ORAC and IA potential were displayed as box plots.

The relationship between AAs and chemical compound composition was first analyzed using Spearman correlation coefficient. Here, R < 0.2 indicates a very weak correlation, R = 0.2–0.4 indicates a weak correlation, R = 0.4−0.6 indicates a moderate correlation, R = 0.6–0.8 indicates a strong correlation and R = 0.8−1.0 indicates a very strong correlation. Likewise, decreasing negative values indicate a stronger inverse correlation. For each AA method (FRAP, ORAC and IA potential), the hop samples were divided into three classes (low, medium and high) regarding Q1 and Q3 of individual AA as the limits among classes (low denotes AA values ≤ Q1, medium denotes AA values between Q1 and Q3 and high denotes AA values ≥ Q3). Within each AA method, ANOVA and post hoc Duncan’s multiple range test were used to test if AA classes (low, medium, high) differ in their chemical constituents. Additionally, a total AA class was constructed based on all three AAs. For this reason, individual AAs were standardized in a continuous space using the equation (x-mean)/sd. Standardized values for each AA method were further summed up for each hop sample. The newly created variable total AA was then divided into three total AA classes, respecting −0.5 and 0.5 as the class limits. Again, ANOVA and Duncan’s multiple range tests were used to test potential differences in chemical constituents among total AA classes.

Principal component analysis (PCA) was performed to calculate the unbiased weight of the components based on their factor scores and the proportion of variance in each component. For this purpose, the rotation method based on Promax with Kaiser normalization was applied for HAE samples.

## 4. Conclusions

To sum up, the determined AAs of HAE extracts represent a promising potential for the use of natural hop extracts for food, feed, pharmaceutical and medicinal purposes. The FRAP values of the different hop genotypes ranged from 63.5 to 101.6 μmol TE/g DW, the ORAC values ranged from 1069 to 1910 μmol TE/g DW and IA potential values ranged from 52.7 to 118.0 mmol TE/g DW. Significant differences in AA were found between different hop genotypes regardless of the AA method used. Therefore, it is also important to know which hop genotype is applied in the brewing process so it can exhibit its antioxidant potential and keep the beer stable during aging.

The results indicated that the individual identified hop compounds do not have a major effect on the AA values determined using individual methods, as well as on the total AA values of the studied hop genotypes. While the ORAC and IA potential methods tend to correlate in statistical tests (Spearman, PCA), the biological significance of the FRAP results may be questionable. Based on the results of this study, as well as previous studies, it could be hypothesized that the synergistic or antagonistic effects have a greater impact on the AA of hop extract than the presence and the quantity of individual hop compounds. 

To conclude, due to their high AA, hop extracts find applications in the food, cosmetic, nutraceutical and pharmaceutical industries by facilitating the formulation of various products beyond their traditional use. Because of the various natural products present in hop extracts, further studies should be oriented to investigate the interactions between them and the remaining components of the matrix in which they are incorporated. Bioavailability should also be thoroughly examined in further studies if hop extracts are ingested as a part of a new product.

## Figures and Tables

**Figure 1 plants-12-03436-f001:**
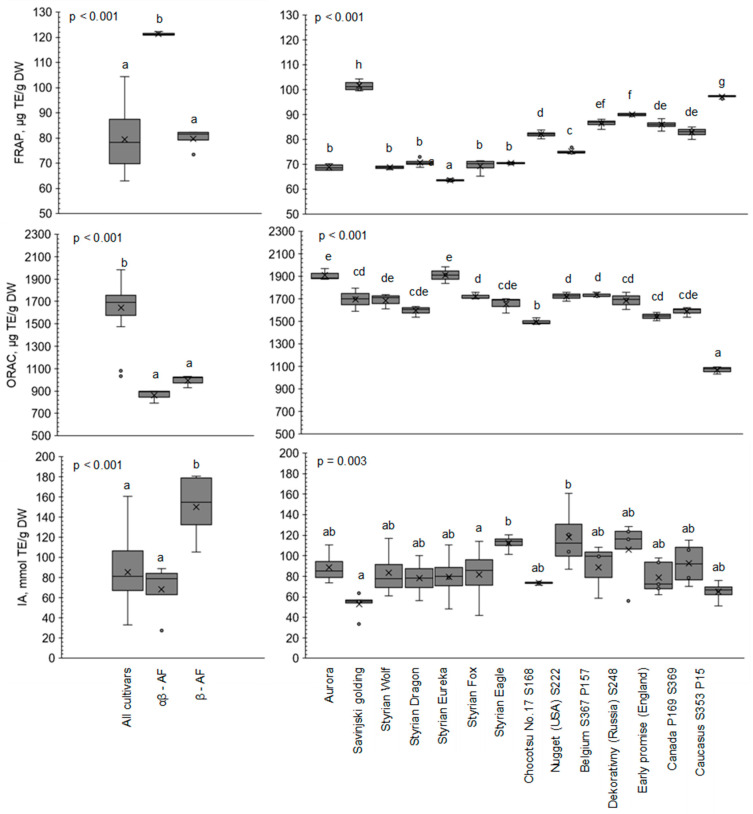
FRAP, ORAC and IA potential values of AA determination. Different superscripts for the hop sample denote significant differences at *p* < 0.05.

**Figure 2 plants-12-03436-f002:**
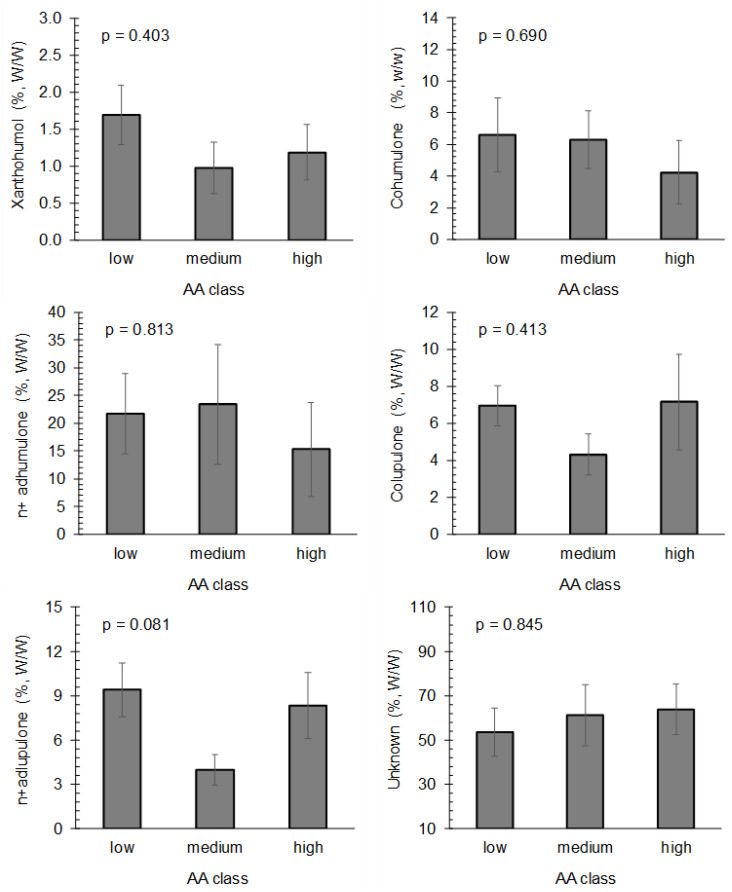
Chemical compound contents according to total AA classes.

**Figure 3 plants-12-03436-f003:**
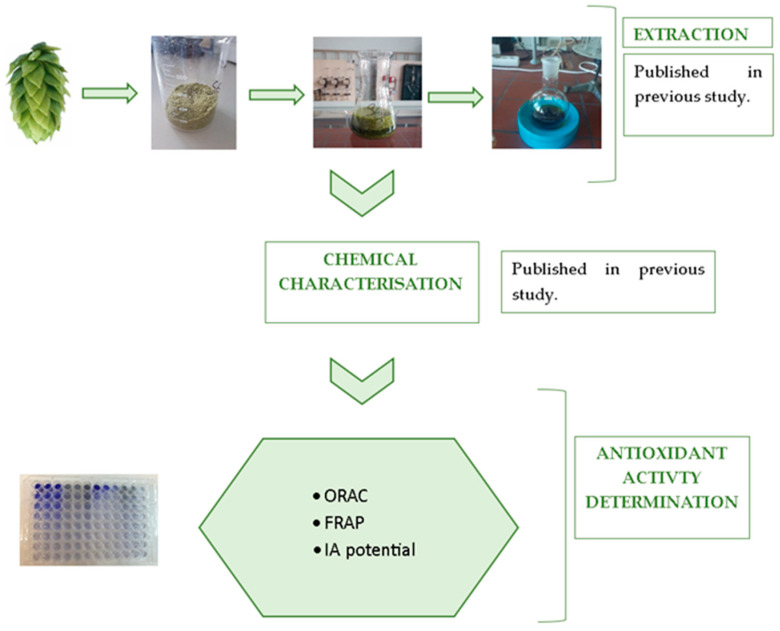
Experimental workflow [27].

**Table 1 plants-12-03436-t001:** Spearman’s rank correlation coefficients (R) between different methods for antioxidative property determination of hop extracts.

Parameter	FRAP	ORAC	IA
FRAP	1.000	−0.511 *	0.359
ORAC	−0.511 *	1.000	0.227
IA	0.359	0.227	1.000

* Correlation is significant at the 0.05 level.

**Table 2 plants-12-03436-t002:** Spearman’s rank correlation coefficients (R) between methods for AA determination of hop extracts and their chemical composition.

Chemical Components/Method	FRAP	ORAC	IA
Xanthohumol	−0.356	0.434	0.044
Cohumulone	−0.121	0.124	−0.129
n+adhumulone	−0.168	0.102	−0.268
Colupulone	−0.263	−0.166	0.137
n+adlupulone	−0.194	−0.184	0.159
Unknown	0.656	0.175	0.141
α-acids *	−0.159	0.066	−0.271
β-acids **	−0.215	−0.219	0.109
Cohumulone and colupulone	−0.271	−0.127	0.112
n+adhumulone and n+adlupulone	−0.129	−0.119	−0.176

* α-acids means cohumulone and n+adhumulone, ** β-acids means colupulone and n+adlupulone.

**Table 3 plants-12-03436-t003:** Chemical compound contents according to AA class using different methods: FRAP, ORAC, IA.

				AA–FRAP						
	Low			Medium			High			*p*
	n	Mean	se	n	Mean	se	n	Mean	se	
Xanthohumol (%, *w*/*w*)	4	1.87	0.55	8	1.23	0.27	4	0.74	0.26	0.179
Cohumulone (%, *w*/*w*)	4	9.06	2.69	8	4.30	1.17	4	5.31	2.59	0.232
n+adhumulone (%, *w*/*w*)	4	28.45	8.65	8	13.91	5.22	4	25.01	16.04	0.469
Colupulone (%, *w*/*w*)	4	6.38	1.95	8	6.85	1.65	4	3.97	0.46	0.498
n+adlupulone (%, *w*/*w*)	4	6.17	1.81	8	8.59	1.93	4	4.76	0.18	0.354
Unknown, %	4	48.1	15.6	8	65.1	7.4	4	60.2	18.7	0.620
				**AA–ORAC**						
	**low**			**medium**			**high**			** *p* **
	**n**	**Mean**	**se**	**n**	**Mean**	**se**	**n**	**Mean**	**se**	
Xanthohumol (%, *w*/*w*)	4	0.74	0.52	8	1.44	0.25	4	1.45	0.48	0.397
Cohumulone (%, *w*/*w*)	4	4.37	2.85	8	6.79	1.56	4	5.03	2.12	0.658
n+adhumulone (%, *w*/*w*)	4	22.59	16.85	8	21.50	6.21	4	15.70	6.54	0.883
Colupulone (%, *w*/*w*)	4	8.13	3.06	8	5.59	0.88	4	4.75	1.88	0.451
n+adlupulone (%, *w*/*w*)	4	9.47	2.87	8	6.81	1.40	4	5.01	1.84	0.377
Unknown, %	4	54.7	17.0	8	57.9	9.6	4	68.1	12.4	0.782
				**AA–IA**						
	**low**			**medium**			**high**			** *p* **
	**n**	**Mean**	**se**	**n**	**Mean**	**se**	**n**	**Mean**	**se**	
Xanthohumol (%, *w*/*w*)	4	1.03	0.46	8	1.52	0.31	4	0.99	0.41	0.528
Cohumulone (%, *w*/*w*)	4	5.55	2.49	8	6.72	1.54	4	4.00	2.55	0.646
n+adhumulone (%, *w*/*w*)	4	26.04	15.66	8	19.91	5.35	4	15.42	10.89	0.784
Colupulone (%, *w*/*w*)	4	5.06	0.91	8	5.88	1.11	4	7.23	3.35	0.749
n+adlupulone (%, *w*/*w*)	4	6.97	2.21	8	6.39	1.48	4	8.35	2.88	0.793
Unknown, %	4	55.4	17.2	8	59.6	9.1	4	64.0	14.8	0.914

**Table 4 plants-12-03436-t004:** List of retained PC weights of all hop chemical compounds and AA determination methods (pattern matrix).

Hop Chemical Compounds/AA Method	PC 1	PC 2
Xanthohumol	**0.907**	−0.004
Cohumulone	**0.819**	0.241
n+adhumulone	**0.843**	0.221
Colupulone	**0.993**	−0.191
n+adlupulone	**0.882**	−0.348
Unknown	**−0.948**	−0.099
FRAP	−0.208	**−0.745**
ORAC	−0.048	**0.814**
IA	−0.199	**0.768**

## Data Availability

All data generated or analyzed during this study are included in this published article.

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
