# Peer review of "Antioxidant Activity of Different Hop (Humulus lupulus L.) Genotypes"

_plants, 2023, doi:10.3390/plants12193436_

Round 1

Reviewer 1 Report

Manuscript ID: plants-2621186

Type: Article

Title: Antioxidant Activity of Different Hop (Humulus lupulus L.)  Genotypes

Authors: Zala Kolenc, Tamara Hribernik, Tomaž Langerholc, Maša Pintarič, Maja Prevolnik Povše, Urban Bren *

Recommendation: Minor Revision

The article “Antioxidant Activity of Different Hop (Humulus lupulus L.)  Genotypes written by Zala Kolenc have been reviewed. Here, the antioxidant activity (AA) of hop extracts obtained from different hop genotypes (n=14) was determined by different methods. The manuscript is well-written and well-planned.  Overall, the manuscript is good and very suitable for the journal.

1.     The title is eye-catching, interesting, and very related to the special issue theme of the journal.

2.     This article is written very well and may be helpful for readers especially those who are working in the area of phytochemistry, or chemistry/biology related to phytochemical compounds around the globe.

3.     The abstract is good and to the point.

4.     The introduction section is written very well. However, there is a lack of literature regarding the metabolomics/proteomics-based mechanistic insights into understanding the phytocompounds. Since current research is mainly focused on the omics era. I am not discouraging the authors, but, the addition of a few sentences would be more helpful to readers.

5.     The material and methods section has been described in detail.

6.     Results and discussion are appropriately discussed. The overall manuscript is good. However, some minor errors have been observed which need to be improved before publication of the article. 

7.     I applaud the authors for Tables and Figures. A little bit more attention would make these more useful for the readers.

8.     Conclusion section is good. However, I think it needs revision/modifications.

Reviewer 2 Report

1. Please state the traditional use of this plant

2. Please provide the p valuse in the result section 

3. Please provide the information on when and where the hop was collected and who identified the plant material.

4. Please provide the source of the Macrophage cell line source and the cells passage number

5. Some methods without proper citation 

6. Please state clearly the suggestion for future studies based on current findings. 

Overall good

Reviewer 3 Report

The authors are encouraged to explain why they used in this manuscript the same chemically standardized plant extracts used in a previous study (Plants 2023, 12, 120. https://doi.org/10.3390/plants12010120). They could have presented both activities (antioxidant and antimicrobial) in one single manuscript. MPDI does not restrict the works for their length.

The aim of the study is not clear. 

Which was the positive control used in the antioxidant assays?

Cytotoxicity assay. The results are not shown in the manuscript. Which is the positive control in this assay?

One single concentration of plant extracts was used to assess the antioxidant effect. Can the authors conclude a possible antioxidant effect?

The conclusion section should be rewritten. In its current form, this section seems a results section.

No comments

Round 2

Reviewer 3 Report

The authors could have presented the results with the same plant extracts in one single manuscript. Instead, they decided to split the information. 

No comments